# Training Sequence Labeling Models using Prior Knowledge

## Abstract

Sequence labeling task (part-of-speech tagging, named entity recognition) is one of the most common in NLP. At different times, the following architectures were used to solve it: CRF, BiLSTM, BERT (in chronological order). The combined model BiLSTM / BERT + CRF, where the last one is the topmost layer, however, performs better than just BiLSTM / BERT. It is common when there is a small amount of labeled data available for the task. Hence it is difficult to train a model with good generalizing capability, so one has to resort to semi-supervised learning approaches. One of them is called pseudo-labeling, the gist of what is increasing the training samples with unlabeled data, but it cannot be used alongside with the CRF layer, as this layer simulates the probability distribution of the entire sequence, not of individual tokens. In this paper, we propose an alternative to the CRF layer — the Prior Knowledge Layer, that allows one to obtain probability distributions of each token and also takes into account prior knowledge concerned the structure of label sequences.

## 1 Introduction

Sequence labeling, along with the text classification, is one of the most common in natural language processing (NLP). In general, the task is to match each token — word or sub-word in a sentence — with a corresponding label. Particular cases of this problem are part-of-speech tagging (POS tagging) and named entity recognition (NER).

Before the rise in popularity of deep learning methods, the sequence labeling problem was solved using graph probabilistic models, among which the Conditional Random Fields (CRF) (Lafferty et al., 2001) architecture showed the best quality. Now, to tackle this problem, both Recurrent Neural Networks (RNN) (Rumelhart et al., 1986) (Karpathy, 2015), for example Bidirectional Long Short-Term Memory (BiLSTM) (Hochreiter & Schmidhuber, 1997) (Schuster & Paliwal, 1997), and neural networks based on the Transformer (Vaswani et al., 2017) architecture, such as Bidirectional Encoder Representations from Transformers (BERT) (Devlin et al., 2018) were used.

The combined model BiLSTM / BERT + CRF (Huang et al., 2015) (Lample et al., 2016), where the last one is the topmost layer, performs better than just BiLSTM / BERT, demonstrating quality close to state-of-the-art. This fact could be explained that CRF layer models a joint probability distribution over the entire sequence, which allows one to take into account the structure of the label sequences, for example, the presence of forbidden subsequences of labels.

It is common when there is a small amount of labeled data, and therefore it becomes challenging to train a neural network with good generalizing capability and one has to resort to semi-supervised learning approaches. One of them is called pseudo-labeling (Lee, 2013), the gist of what is increasing the training samples with unlabeled data. To implement this approach, it is required that the model returns a probability distribution over each token in the sequence, but when using CRF, this cannot be done, since this layer models the probability distribution over the entire sequence.

To achieve the desired result, BiLSTM / BERT model without the CRF layer could be used, but the structure of the label sequences will not be taken into account, which leads to worse quality of the model.

In this paper, we propose an alternative to the CRF layer — the Prior Knowledge Layer (PKL), that allows one to obtain probability distributions of each token through a baseline model such as

BiLSTM / BERT, and also takes into account prior knowledge of the structure of the label sequences, which makes it possible to use this architecture together with pseudo-labeling approach.

## 2 BACKGROUND

### 2.1 SEQUENCE LABELING MODELS

The problem of sequence labeling is widely known and there are many ways to approach it. One of the first models used to solve this is the Hidden Markov Model (HMM), which has its origins in Leonard Baum's 1966 work (Baum & Petrie, 1966) (Rabiner & Juang, 1986). This model has found its application not only in sequence labeling, but also in speech recognition and analysis of biological sequences, in particular DNA. This model was replaced with the Maximum-Entropy Markov Model (MEMM, 2000) (McCallum et al., 2000), which combines the features of the HMM and Maximum Entropy models. A breakthrough was marked with the appearance of the Conditional Random Fields (CRF, 2001) model, which at that time demonstrated state-of-the-art quality. There are also modifications and generalizations of CRF, such as CRF with partial training (Mann & McCallum, 2007), Hidden-state CRF (HCRF) (Quattoni et al., 2007), Dynamic CRF (DCRF) (Sutton et al., 2007) and Continuous CRF (CCRF) (Qin et al., 2008).

The disadvantage of the aforementioned model is the need for manual feature generation and domain knowledge to train a model with good generalizing capability. This problem was solved by deep neural networks, which do not require manual generation of features, and also show better results, provided that there is a sufficient amount of labeled data.

Based on the sequence labeling problem structure, which implies a sequence at the input and a sequence of the same length at the output, the architecture of the Recurrent Neural Network is well-suited for its solution. RNNs are known to have a vanishing gradient problem, so the other architectures such as Long Short-Term Memory (LSTM, 1997) (Hochreiter & Schmidhuber, 1997) and Gated Recurrent Unit (GRU, 2004) (Cho et al., 2014) (Chung et al., 2014) have been invented to solve this problem.

Sequence labeling task allows reading the entire input sequence to make a decision, that makes it possible to use bidirectional architectures, particularly Bidirectional RNN (BiRNN) (Schuster & Paliwal, 1997), to process the input sequence in two directions from left to right and from right to left for better performance. But there is an important point, that such architectures are not "true" bidirectional, but rather two unidirectional ones.

Thus, we come to one of the most commonly used models for solving the problem of sequence labeling, namely Bidirectional Long Short-Term Memory (BiLSTM) (Hochreiter & Schmidhuber, 1997) (Schuster & Paliwal, 1997).

In 2017, as the article "Attention Is All You Need" (Vaswani et al., 2017) was published, in which the authors presented a completely new architecture called Transformer. This architecture surpassed all previous state-of-the-art models in the task of machine translation. The authors of the article continued the ideas of the seq2seq (Cho et al., 2014) (Sutskever et al., 2014) models and the attention mechanism (Bahdanau et al., 2014) introducing two new operations: Scaled Dot-Product Attention and Multi-Head Attention. These operations allow to consider information about all tokens in the sentence when processing a specific token, which makes them "truly" bidirectional.

Transformer is an encoder-decoder model, and therefore cannot be used for sequence labeling. But it was just a matter of time before the bidirectional attention mechanism and Transformer architecture was adapted in other NLP tasks, and in 2018 the Bidirectional Encoder Representations from Transformers (BERT) (Devlin et al., 2018) model appeared. Architecturally, BERT is a Transformer encoder, and this model can be used to solve a wide range of NLP tasks, including sequence labeling.

The next architectural improvement was to combine the baseline model like BiLSTM / BERT with CRF (Huang et al., 2015) (Lample et al., 2016), where the last acts as the topmost layer and models the joint probability distribution over the entire sequence. In this approach, the need for manual generation of features becomes irrelevant, since the logits of the baseline model, which are used as features, enter the CRF. BERT + CRF model shows quality close to state-of-the-art.

## 2.2 Pseudo-Labeling

Pseudo-labeling is a simple and effective semi-supervised method, that was originally invented for deep neural networks. The core of it is that the model is jointly trained using both labeled and unlabeled data. First, model trained on labeled data, and then used for marking unlabeled data — taking label with the maximum predicted probability given by the model for each data sample. Then these labels used as true labels for fitting the model on all data (Lee, 2013).

## 2.3 Named Entity Recognition tagging

Named Entity Recognition task consists in determining and extracting subsequences of words, called spans, related to one named entity. In order to classify entire spans as named entities, special markups (Carpenter, 2009) are used. The most popular markups are BIO (Ramshaw & Marcus, 1995) and BILUO (BIOES, BMEWO) (Borthwick, 1999).

Name BIO stands for Beginning-Inside–Outside. Within this markup, there are tags such as named entity names and O (Outside), followed by one of the two prefixes B- (Beginning), I- (Inside). The B-prefix corresponds to the beginning of the named entity, the I- prefix corresponds to the continuation of the named entity, in case if the named entity consists of more than one token.

BILUO markup extends BIO with two additional prefixes L- (Last), U- (Unit), which denote the end of a named entity, in case the named entity consists of more than one token, and a single named entity, consisting of one token, respectively.

Note that after applying one of these markups, the number of labels becomes greater than the number of named entities. Let the number of named entities be $N$, then:

- the number of labels in the BIO markup is $2N + 1$
- the number of labels in the BILUO markup is $4N + 1$

Despite the increase in the number of labels, not every label can follow arbitrary other label. Let's denote the named entity by $X$.
BIO markup:

- the label I-$X$ can only follow the label B-$X$

BILUO markup:

- the labels I-$X$ and L-$X$ can only follow the label B-$X$
- label O can only follow label O, L-$X$ or U-$X$

In other words, there are pairs of consecutive labels that are not correct and contradict the rules described above.

It is important to note that the structure of label sequences can be in any sequence labeling task, but this structure is most indicative in NER, namely BIO / BILUO markups.

## 3 Method

### 3.1 Problem statement

Let $\mathbb{D}$ be the set of labeled sequences $(x, y)$, where:

- x = $(x_1, ..., x_l)$ - sequence of tokens from $\mathbb{X}$
- y = $(y_1, ..., y_l)$ is a sequence of labels from $\mathbb{Y}$

In the sequence $(x, y) \in \mathbb{D}$, the token $x_i$ corresponds to the label $y_i$.

Consider an arbitrary probabilistic model with parameters $\theta$: $p_\theta(y|x)$
The Maximum Likelihood principle for solving the sequence labeling problem looks as follows:

$$\sum_{(x,y)\in\mathbb{D}} \ln p_\theta(y|x) \to \max_\theta$$

Optimal sequence of labels with known parameters $\theta$:

$$\hat{y} = \arg\max_{y \in \mathbb{Y}^l} p_\theta(y|x)$$

## 3.2 PRIOR KNOWLEDGE MATRIX

For a convenient representation of correct / incorrect pairs of consecutive labels, it is convenient to use a matrix, which we call the Prior Knowledge Matrix (PKM). The rows of the matrix will correspond to the labels at the $i$ position, and the columns will correspond to the labels at the $i + 1$ position. Let 1 be an incorrect pair of labels, and 0 a correct one.

This approach is generalized to correct / incorrect sequences of labels of length more than two by transition from matrices to multidimensional arrays (tensors) of dimension, corresponding to the length of the largest forbidden sequence. The number of incorrect label sequences increases exponentially with the length of the label sequence, while the number of correct sequences increases much more slowly. This leads to the fact that the resulting multidimensional array is very sparse.

## 3.3 PRIOR KNOWLEDGE MATRIX FOR NER

Consider the NER task with two named entities:

- person (PER)
- location (LOC)

Let's build a PKM for both BIO and BILUO markups.

PKM row/column names for:

- **BIO**: *O*, *B-PER*, *B-LOC*, *I-PER*, *I-LOC*
- **BILUO**: *O*, *B-PER*, *B-LOC*, *I-PER*, *I-LOC*, *L-PER*, *L-LOC*, *U-PER*, *U-LOC*

$$\begin{pmatrix} 0 & 0 & 0 & 1 & 1 \\ 0 & 0 & 0 & 0 & 1 \\ 0 & 0 & 0 & 1 & 0 \\ 0 & 0 & 0 & 0 & 1 \\ 0 & 0 & 0 & 1 & 0 \end{pmatrix}$$

Figure 1: Prior Knowledge Matrix for BIO tagging

$$\begin{pmatrix} 0 & 0 & 0 & 1 & 1 & 1 & 1 & 0 & 0 \\ 1 & 0 & 0 & 0 & 1 & 0 & 1 & 1 & 1 \\ 1 & 0 & 0 & 1 & 0 & 1 & 0 & 1 & 1 \\ 1 & 1 & 1 & 0 & 1 & 0 & 1 & 1 & 1 \\ 1 & 1 & 1 & 1 & 0 & 1 & 0 & 1 & 1 \\ 0 & 0 & 0 & 1 & 1 & 1 & 1 & 0 & 0 \\ 0 & 0 & 0 & 1 & 1 & 1 & 1 & 0 & 0 \\ 0 & 0 & 0 & 1 & 1 & 1 & 1 & 0 & 0 \\ 0 & 0 & 0 & 1 & 1 & 1 & 1 & 0 & 0 \end{pmatrix}$$

Figure 2: Prior Knowledge Matrix for BILUO tagging

## 3.4 PRIOR KNOWLEDGE LAYER

Let's take a look at how the Prior Knowledge Layer (PKL) works. Since this layer takes into account knowledge about incorrect label subsequences, this layer initializes with Prior Knowledge Matrix.

PKL has no trainable parameters. It receives as input the logits of each token obtained by the baseline BiLSTM / BERT model and returns a scalar loss $\alpha$ — PKL loss — that acts as an weighted additive term in the minimized loss function. Let's convert the Maximum Likelihood principle to minimization optimization problem multiplying by -1 and add scalar additive term:

$$-\sum_{(x,y)\in D} \ln p_\theta(y|x) + \lambda\alpha \to \min_\theta$$

$\lambda$ acts as a regularization parameter and could be tuned as hyperparameter.

PKL loss is equal to the sum of all scalar penalties, calculated on all consecutive pairs (in the general case, sequences of arbitrary length) tokens, taking into account an information of incorrect labels subsequences - Prior Knowledge Matrix. To calculate one scalar penalties, we need to define an operation on two probability distributions (over two consecutive tokens) and one matrix (PKM), which will penalize model for high probability values corresponding to incorrect pairs of labels.

Let's denote the probability distribution matrices, corresponding to each token in consecutive pair as $P_1$ and $P_2$, correspondingly and Prior Knowledge Matrix as $K$.

One way to specify such an operation is the average over diagonal elements of the product of the following three matrices $P_1$, $P_2$ and $K$:

$$mean(diag(P_1 K P_2))$$

Since the loss function additive term, provided by Prior Knowledge Layer, takes into account information about incorrect label subsequences, this leads to:

- better generalization capability, especially with a small amount of training data
- faster convergence, comparing with baseline model

Prior Knowledge Layer uses only during training phase. During inference the model behaves the same as the baseline BiLSTM / BERT model without CRF layer.

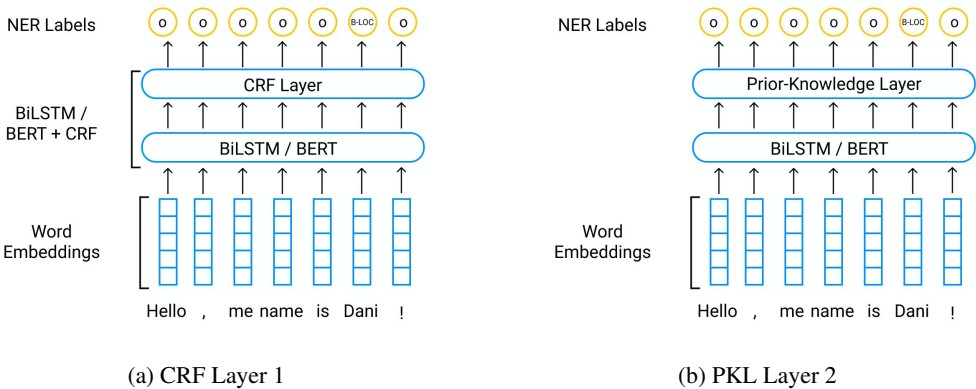

(a) CRF Layer 1        (b) PKL Layer 2

Figure 3: Baseline models with different topmost layers

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
