# OpenReview forum: "Training sequence labeling models using prior knowledge"
_ICLR.cc/2022/Conference — ICLR 2022 Submitted_

### Official Review · Reviewer_UZd9 · 2021-10-29

**Correctness:** 4
**Technical Novelty And Significance:** 1
**Empirical Novelty And Significance:** Not applicable
**Recommendation:** 1
**Confidence:** 5

**Details Of Ethics Concerns:**

Nil

**Main Review:**

Strengths:
- The authors addressed state transition problems in sequence labeling problems, and proposed an approach that seems sound. This could be useful in semi-supervised learning scenarios when sequences are sparsely labeled at the word level.

Weaknesses:
- There is no experimental results that validates the proposed approach. The authors should conduct experiments to show that their approach outperforms CRF in the semi-supervised scenarios they proposed.
- The authors should compare their approach with CRF with partially labeled sequences, e.g., [1].

[1] Tsuboi, Yuta, et al. "Training conditional random fields using incomplete annotations." Proceedings of the 22nd International Conference on Computational Linguistics (Coling 2008). 2008.

**Summary Of The Paper:**

The paper formulated an approach to impose valid state transitions from t to t+1 for sequence labeling problems. The proposed approach replaces the usual conditional random fields with what is named as a prior knowledge layer (PKL), which simply multiplies a state transition matrix which zeros invalid transition probabilities.


**Summary Of The Review:**

I would recommend to reject this paper as the theoretical novelty is insufficient, and there are no empirical results.

---

### Official Review · Reviewer_n5Vw · 2021-10-31

**Correctness:** 1
**Technical Novelty And Significance:** 2
**Empirical Novelty And Significance:** 1
**Recommendation:** 1
**Confidence:** 5

**Main Review:**

Strengths:
- this paper attempts to address an important problem in sequence labelling of data sparsity, as well as a weakness of CRF classifiers.

Weaknesses:
- The technical and experimental rigour of this paper is critically lacking.
- The paper has no experimental section - no benchmarks vs current state of the art, no ablations for the PKM layer, and no validation of any of the claims made about the effectiveness of the PKM layer.
- For example, these claims are left completely unvalidated:
-- The number of incorrect label sequences increases exponentially with the length of the label sequence, while the number of correct sequences increases much more slowly. This leads to the fact that the resulting multidimensional array is very sparse.
-- better generalization capability, especially with a small amount of training data
-- faster convergence, comparing with baseline model
- the presentation and communication is also unclear in various areas, including when describing the current literature (e.g. unclear what tasks are being considered for sequence labelling)

**Summary Of The Paper:**

The paper proposes a new type of classification layer, called the prior knowledge layer (PKL) that estimates the probability distribution of tokens and can incorporate prior knowledge about the prior knowledge of the structure of label sequences. The authors do not validate any claims made in the paper with experimental results, benchmarks, or datasets.

**Summary Of The Review:**

Overall, the paper critically lacks any experimental rigour so it is impossible to evaluate any of the technical claims made.

---

### Official Review · Reviewer_7Kg2 · 2021-11-02

**Correctness:** 3
**Technical Novelty And Significance:** 1
**Empirical Novelty And Significance:** 1
**Recommendation:** 1
**Confidence:** 4

**Main Review:**

I'm afraid that the idea of this paper is too simple. In fact, utilizing prior knowledge of BIO tagging scheme for safe decoding is very prevalent in practice. For example, I-ORG must follow B-ORG rather than O. Besides, no experiments are found in the paper. The whole paper just consists of 6 pages. It seems that the authors have submitted an incomplete version.

**Summary Of The Paper:**

This paper proposes a prior knowledge matrix to displace CRF in sequence labeling. The author also claims that this approach is useful for low-resource settings, especially considering using pseudo-labeling.

**Summary Of The Review:**

1, too simple idea, even not a novel one
2, no experiments

---

### Official Review · Reviewer_E7aG · 2021-11-05

**Correctness:** 2
**Technical Novelty And Significance:** 1
**Empirical Novelty And Significance:** 1
**Recommendation:** 1
**Confidence:** 5

**Main Review:**

This is not a paper, instead technically more like a toy report. The architecture of the entire manuscript is way incomplete, without enough technique detail, not to mention the lost experiment part. Also, the idea of prior knowledge integration is naive.

**Summary Of The Paper:**

The authors try to propose a `Prior Knowledge Layer' as an alternative to the CRF layer for the standard sequence labeling framework in NLP.

**Summary Of The Review:**

Introduction: without a motivation; Related work: not informative; Method: no detail & incomplete; Experiment: missing. Reference: bad formation. So I would recommend a rejection.

---

### Decision · Program_Chairs · 2022-01-20

**Decision:**

Reject

**Comment:**

This paper presents an approach for using prior knowledge to constrain transitions for consecutive time steps and aims to replace conditional random fields for sequence tagging tasks in sequence labeling. However, the paper seems incomplete with no experimental results and analysis to validate the proposed ideas.